# Different Translational Activities of Inflammatory Regulators Associated with Hypervolemia in Haemodialysis Patients

**DOI:** 10.3390/ijms26188922

**Published:** 2025-09-13

**Authors:** Christof Ulrich, Amanda Dawood, Roman Fiedler, Silke Markau, Matthias Girndt

**Affiliations:** 1Department of Internal Medicine II, Martin Luther University Halle-Wittenberg, 06120 Halle, Germany; amanda.dawood@student.uni-halle.de (A.D.); roman.fiedler@uk-halle.de (R.F.); silke.markau@uk-halle.de (S.M.); matthias.girndt@uk-halle.de (M.G.); 2KfH Nierenzentrum Halle (Saale), 06120 Halle, Germany

**Keywords:** haemodialysis, hypervolemia, polysome profiling, PBMCs, IL-10, TIPE2, OTUD1

## Abstract

The inflammatory burden in chronic kidney disease patients under maintenance haemodialysis (HD) is high. Overhydration (hypervolemia) associates closely with inflammation in HD, while its patho-mechanisms are still elusive. Anti-inflammatory mediators such as IL-10 and TIPE2, as well as anti-inflammatory regulators such as OTUD1, are key controllers of cellular homeostasis after chronic inflammatory insults. Forty-two HD patients and nine healthy controls (CO) were enrolled in a cross-sectional pilot study. Bioimpedance measurements were performed to dichotomise the HD patients into a normovolemic (N) and a hypervolemic (H) group. Polysome profiling (inclusive monosomal, early and late polysomal peak analysis) followed by translational activity analysis of IL-10, TIPE2 and OTUD1 were performed. mRNA expression and translational activity of IL-10 were neither different between N and H nor between HD and CO. Significantly higher TIPE2 mRNA expression in PBMCs was measured in H versus CO, whereas translational activity failed to be different in the three cohorts at all. In contrast, monosomal translational activity of OTUD1 was significantly different when H vs. N and H vs. CO were compared. Inflammation is insufficiently balanced in chronic kidney disease, as the translational activity of the anti-inflammatory mediators IL-10 and TIPE2 are not different either between N and H or between HD and CO. The deubiquitinase OTUD1 appears to be significantly upregulated in hypervolemia. As not only TIPE2 but also different other molecules are target of OTUD1, the exact mechanistic interaction between OTUD1 and TIPE2 in HD must be evaluated.

## 1. Introduction

Hypervolemia can be characterised by an excessive accumulation of fluid in the extracellular space of the body. Depending on the degree of overhydration oedema, dyspnoea, weight gain, fatigue, increased blood pressure and decreased urine output can be typical symptoms of fluid overload [1,2]. The causes of underlying hypervolemia are multifaceted. Congestive heart failure, liver cirrhosis, excessive sodium intake, anti-inflammatory drugs intake (NSAID) and kidney dysfunction may be associated with hypervolemia [3]. Fluid overload is common haemodialysis patients (HDs) and mostly caused by excessive fluid intake and heart failure coupled with decreased urine output [4]. But without doubt inflammation is part of overhydration in HD patients, too. Several inflammatory markers, including CRP, TNF-α and IL-6 are increased in hypervolemic HD patients [5,6,7], and NF-kB, a central transcription factor mediating inflammation, appears to be activated in many nephropathic diseases [8].

In “healthy people”, inflammatory insults are always accompanied by anti-inflammatory counteractive measures. Among the plethora of inflammatory and anti-inflammatory molecules IL-10, TIPE2 and OTUD1 may exert significant influence on anti-inflammatory signalling and its regulation [9,10,11].

Compensation of a proinflammatory state in patients with chronic kidney diseases seems to be necessary. But inflammatory effects exerted by chronic uremic intoxication and hypervolemia can only partly be balanced in these patients. Recently we showed that although IL-10 mRNA was increased in hypervolemic patients, translation of the anti-inflammatory molecule appeared to be insufficient [7]. IL-10 is an important counter-regulatory mechanism dampening inflammation. This is true in intestinal epithelial cells to regulate gut homeostasis and cells in the arterial cell wall to restrict cell wall inflammation, as well as in bacterial and toxin-induced diseases to fight against an inflammatory cytokine storm [12,13,14]. To decode the disproportion of IL-10 transcription and translation, the technique of polysome profiling may give an answer, as only IL-10 molecules attached to ribosomes will be detected by this technique [15,16].

Further, it has been shown that RelB activation appears to be increased in hypervolemic HD patients [17]. Indeed, anti-inflammatory mechanisms that take part in the regulation of the Rel/NF-kB family seem to be of particular interest in kidney failure patients [18]. In resting cells, Rel-dimers remain inactive, coupled to the inhibitory subunit IkB which can be phosphorylated, ubiquitylated or finally proteasomally degraded upon stimulation with LPS, TNF or other different stimuli [19]. This turns our attention to putative proteins and mechanisms that regulate NF-kB activation. Among them are TIPE2 (TNFAIP8L2), a member of the tumour necrosis factor alpha-induced proteins, and OTUD1, belonging to the OTU deubiquitinase family [20]. TIPE2 is a negative regulator of inflammation and lipid biosynthesis and is exclusively expressed in myeloid cells [21,22,23]. Furthermore, TIPE2 may negatively regulate T-cell activation—the T-cell activation receptor (TCR) CD69 and the cytokine interferon-γ are reduced in cells overexpressing TIPE2 [23]. In macrophages, there is evidence that TIPE2 deficiency is linked to toll-like receptor (TLR) hyperresponsiveness that makes the cells more sensitive to LPS. The impact of TIPE2 on inflammation is also strengthened by data showing elevated levels of IL-6 in TIPE2-knockout mice [23]. Thus, it is suggested that TIPE2 is an immune checkpoint molecule which negatively regulates TLR and TCR function, preventing hyperresponsiveness and maintaining immune homeostasis [22,23].

Ubiquitination is a reversible post-translational modification that activates different molecules [20]. By removal of ubiquitin residues, deubiquitinating enzymes (DUBs) deactivate the corresponding molecules by marking them for proteasomal degradation. Thus, the DUB OTUD1 is an important regulator of anti-viral and -inflammatory immune responses [24]. It takes part in canonical NF-kB activity by hydrolysing K63-linked ubiquitin chains of different NF-kB signalling factors such as IRAK1, NEMO and LUBAC. Another interesting aspect regarding this study concerns TIPE2, which also appears to be a target of OTUD1. OTUD1 can deubiquitinate TIPE2 and thus, e.g., negatively regulate inflammatory responses in sepsis-induced lung injury [25].

To effectively determine the translational products of IL-10, TIPE2 and OTUD1, we decided to use polysome profiling. This is a powerful technique for studying translated RNAs and an appropriate means to detect discrepancies between mRNA and protein expression. It is based on sucrose-gradient separation of translated mRNAs that are associated with mono- (one ribosome) or polysomes (two to several ribosomes) [26]. The detection of actively translated mRNAs is of high importance, because almost half of the variation (besides transcriptional, epigenetic effects) is due to translational control [27].

Studying the translational profiles of IL-10, TIPE2 and OTUD1 may shed light on the inflammatory machinery triggered by fluid overload in HD patients.

## 2. Results

### 2.1. Demographic Data

The cross-sectional study included hyper- (H) and normovolemic HD patients (N) and a small cohort of healthy subjects (COs). The H patients were significantly older than the N patients, and the COs were significantly younger than the H patients. Overhydration characteristics like phase angle, resistance, reactance and intracellular (ICW) and extracellular cell water (ECW) significantly differed between both groups. H patients also had significantly higher systolic blood pressure and increased CRP values (Table 1). In contrast to HD patients, the COs were free of signs of inflammation (Table 1).

### 2.2. Leucocyte Numbers

The number of CD45-positive leucocytes in the peripheral blood was not different between normovolemic and hypervolemic HD patients. However, the T-helper cell fraction (CD4+) was significantly lower in H patients (Table 2).

### 2.3. Sequence Analysis

mRNA-sequence analysis from whole blood samples was investigated in HD patients to get an idea which genes might be differently expressed in hyper- and normovolemic patients. RNAs from 10 patients (3 hyper-/7 normovolemic) were sequenced. An exemplary MA diagram (differential expression in a normovolemic patient compared to a hypervolemic counterpart) demonstrates that relatively more genes are upregulated (red dots) than downregulated (blue dots) in N vs. H patients (Figure 1).

About 13,000 different transcripts were found in the whole blood isolates of both groups. Higher FKPM reads were found for 345 transcripts in normovolemic and 33 in hypervolemic patients. Some of the differently regulated genes are listed below (Table 3). Among them, a peptidase (ASPRV1), a proteinase (CTSE), a phospholipase (GPLD1) and a deubiquitinase (OTUD1) were found—all of which were downregulated in normovolemic HD patients. Conversely, the transcription factor SOX8, the MAPK-regulated co-repressor-interacting protein 2 (MCRIP2), cyclin F (CCNF) and TNF-α-induced protein8-like 2 (TNFAIP8L2, TIPE2) were found to be upregulated in N vs. H HD patients.

As we focused our analysis on anti-inflammatory regulatory genes, we had a closer look at IL-10 and TNFAIP family members. Interestingly, in all patients analysed, no transcripts for IL-10, IL-22 or IL-26 could be detected, while several members of the TNFAIP family were found. Among them, A20 and Pentraxin 3, two molecules which are related to inflammation, were not statistically different between N and H patients, and TNFAIP8L2 (TIPE2) expression was significantly enhanced in N but not in H patients (Table 4).

### 2.4. mRNA Expression of TNFAIP and IL-10 Members and OTUD1 in Whole Blood

Next, we confirmed the mRNA-seq screening analysis by qPCR in whole blood samples. With this technique, *TIPE2* gene expression was also shown to be significantly enhanced in N but not in H patients. TNFAIP3 and TNFAIP levels were not different between the groups (Table 5). As expected after mRNA-seq analysis, no members of the IL-10 family tested showed expression differences between N and H patients (Table 5).

### 2.5. Polysome Profiling of IL-10, TIPE2 and OTUD1

Polysome profiling is a powerful tool used to detect the “real” translational activity and remove “empty RNAs” (RNAs not bound to the ribosome) which are also detected by qPCR of mRNA. Representative figures of polysome analysis of a normovolemic, hypervolemic and control subject are depicted in Figure 2.

Typical characteristics such as the 60S ribosome, monosomal and polysomal peaks (differentiated into the early and late polysome) are marked in Figure 2. Regarding mRNA expression of IL-10 (Figure 3a), as well as monosomal (Figure 3b), early polysomal (Figure 3c) and late polysomal (Figure 3d) translational activity, there was no difference between the three groups. In contrast, the mRNA expression of TIPE2 (Figure 3e) was significantly elevated in H patients as compared to healthy control patients. However, this result is misleading, as both the monosomal (Figure 3f) and polysomal translational activities (Figure 3g,h) are not different between the three groups. Elevated mRNA expression of OTUD1 in H patients is detected when hypervolemic and control patients are compared (Figure 3i). This holds true for monosomal translational activity analysis (Figure 3j). Additionally, H patients also have a higher OTUD1 activity level compared to their normovolemic counterparts (Figure 3j). This observation also applies to their early polysomal activity (Figure 3k), while the late polysomal activities were not different between the three groups (Figure 3l).

### 2.6. Multivariate Linear Regression Model

There remains the question of whether the monosomal difference in OTUD1 mRNA expression between N and H patients is dependent on other relevant factors. Approaching this question, we performed a multivariate linear regression analysis using OTUD1 mRNA expression data (monosomal (fraction 5), which is significantly different between N and H patients) and phase angle (determinant of cell integrity and nutritional status) as fixed factors, and age, Kt/V and CRP values as covariates (Table 6).

## 3. Discussion

In this study, we investigated the translational activity of IL-10, TIPE2 and OTUD1, proteins that are involved in the regulation of inflammation in normo- and hypervolemic HD patients. Hypervolemic HD patients, categorised by bioimpedance measurement, have higher intracellular and extracellular water accumulation. In agreement with other studies, we find clinically higher systolic blood pressure and enhanced levels of the inflammation marker CRP in H patients [28,29]. Most interestingly, N and H patients significantly differ in terms of age. As age is a powerful predictor of most diseases, this fact may impact our study setup, including, e.g., phase angle, IL-10 and OTUD1 expression. However, as analysed by multivariate expression, age affects the phase angle rather than OTUD1 mRNA expression. The phase angle reflects cellular health and nutritional status. Thus, low phase angle values coincide with low cellular integrity and worse nutritional status. This appears to make sense for “the older” H patients. In contrast, OTUD1 mRNA expression is dependent on inflammatory status, but not on age or Kt/V.

IL-10 is suggested to be one of the most important anti-inflammatory cytokines in immune cells [9] and therefore of particular interest for balancing overshooting inflammatory reactions and re-introducing tissue homeostasis. In an earlier study, we found elevated mRNA transcription rates in H patients, while protein levels were not different between N and H patients [30]. Thus, the idea was raised that polysome profiling could shed light on this discrepancy. In this study, however, we do not see elevated mRNA expression levels in H compared to N patients. In agreement with this observation are the IL-10 translational activity data as measured by polysome profiling analysis. Maybe the different sources of mRNA in the two studies (whole blood versus PBMCs—granulocytes are also capable of expressing IL-10) are the cause of the different mRNA results. But it is also intriguing that neither IL-10 mRNA nor mRNAs of IL-10 family members have been detected in the 10 samples which have been sequenced. So, as there is no evidence that a slow translational efficacy (monosome versus polysome translation) is evident as an explanation for the low IL-10 protein content in immune cells of H patients, we are confident that our previous suggestion involving insufficient STAT3 phosphorylation and putative degradative mechanisms initiated by miR-142-3p might be causal of the patients’ low IL-10 protein levels [30].

It is of note that the mRNA-seq analysis in a small cohort of normo- and hypervolemic HD patients did not find mRNA expression relevant to IL-10. This may be an indication that hypervolemia does not trigger IL-10 per se. We observed only a minor set of genes being upregulated in hypervolemic patients. About ten-fold more genes appeared to be upregulated in normovolemic patients. These data indeed suggest that the impact of hypervolemia on HD patients is suppressing rather than activating.

TIPE2 mRNA expression analysis may fit into this line of reasoning. The mRNA of this anti-inflammatory molecule was upregulated in normovolemic compared to hypervolemic patients. The regulatory impact of TIPE2 on TLR functions in macrophages is of particular interest. Sun and co-workers impressively demonstrated that TIPE2 deficiency was associated with a highly increased inflammatory cytokine storm when mice underwent low-dose LPS injection. A major signalling pathway of mediators like IL-1 and IL-6 is the NF-kB pathway [23]. It was proven that TIPE2 takes part in the regulation of IkB phosphorylation and NF-kB translocation. Therefore, TIPE2 is an anti-inflammatory mediator—the lower the TIPE2 expression, the higher the sensitivity for TCR and TLR activation [31]. So, it seems plausible that the inflammatory burden in hypervolemic compared to normovolemic patients is higher. Lower TIPE2 mRNA levels in H-HD patients may favour chronic inflammatory conditions and may prevent cellular homeostasis. Thus, other reasons must account for the higher mRNA TIPE2 expression in N patients. However, we must keep in mind that the translational product of TIPE2 mRNA expression is most likely not increased in N patients, as the translational activity of TIPE2 is—according to polysome profiling—not different between the two groups. This may be an impressive example demonstrating that, most obviously, not all the existing mRNA is processed by ribosomes—either mono- or polysomes—for translation.

The most important issue in this study is OTUD1. OTUD1 is “activated” in hypervolemic patients, as its translational activity is significantly increased at the monosomal (translation by one ribosome) and early polysomal (translation by at least two ribosomes) levels compared to normovolemic HD patients. This result also highlights the meaning of translation machinery. Even if mRNAs are expressed differently between two patient groups, efficient monosomal or polysomal translation mostly determines the final amount of the protein. So, results from the sequence analysis demonstrating the differential expression of cells involved in immune function, i.e., SOX8 and MCRIP2 upregulation in N patients and CTSE and GPLD1 in H patients, need to be examined on the protein level.

As TIPE2 is a target of OTUD1 [25], it seems possible that OTUD1 can deubiquitinate activated TIPE2, and initiate proteasomal degradation this way. However, as we found no differences in translational activity between N and H patients, we suggest that OTUD1 does not differently affect TIPE2 in N and H patients. As both mRNA and protein expression are increased in H patients, OTUD1 should impact immunologic and biochemical processes in H patients. It is known that not only different components of the NF-kB pathway [24], but also STAT3 signalling [32] and interferon signalling via TyK2 are influenced by OTUD1 [33]. Regarding interferon type I signalling, OTUD1 negatively regulates this pathway by disrupting ubiquitination of IRF3 [34]. It is suggested that OTUD1 acts as a checkpoint maintaining immune homeostasis. It restricts RIG-I-mediated immune responses and blocks interferon type I responses [35].

Whether TIPE2 and OTUD1 also have an influence on NF-kB signalling cannot be answered by our study. But the observation that TIPE2 expression is not different between N and H patients may be an indication that at least the classical NF-kB pathway is not differently activated in both groups. In contrast, we could demonstrate that the phosphorylation state of the Rel B molecule is different between N and H patients [17]. This, of course, is independent of both TIPE2 and OTUD1 regulation. Regarding other mechanisms which may influence inflammatory regulators, one such mechanism is oxidative stress. It is well known that haemodialysis is associated with increased oxidative stress which cannot adequately be compensated for by the antioxidative capacity (superoxide dismutase) of dialysis patients [36,37]. Furthermore, there is evidence that, for example, serum protein carbonyl, another marker of oxidative stress, is also elevated in overhydrated HD patients [38]. Unfortunately, we did not measure oxidative stress parameters to see if there are any relations between them and IL-10, OTUD1 or TIPE2. Therefore, it is quite possible that oxidative stress affects the anti-inflammatory machinery, most probably including IL-10 and OTUD1. This, however, is a future topic to be studied.

### Study Limitations

We used bioimpedance measurement to determine the hydration status of our HD patients. Validation of these data by other methods such as lung ultrasound or inferior cava diameter assessments were unfortunately not available for this study. Furter on, for the RNA-sequence analysis only samples of 10 patients were available. This indeed could bias the results and may limit the generalizability of the molecular findings. There may also concerns that only 42 HD patients could be enrolled in the study. However, taking in mind that this investigation was planned as a pilot study we are confident that the low patient number is sufficient to obtain some ideas about both the differently expressed genes and the putative role of an important regulator of inflammation, namely OTUD1 in normo- and hypervolemic patients.

## 4. Materials and Methods

### 4.1. Study Population

This cross-sectional pilot study included 9 healthy control (mean age: 39.7 ± 19.4, 33% female) subjects who were recruited from the medical staff of the university hospital and KfH Dialysis Centre of Halle, as well as 42 haemodialysis patients (mean age: 59.6 ± 15.0, 43%female). At first, 81 HD patients were recruited from the KFH Dialysis Centre, which is associated with the Department of Internal Medicine II of University Halle-Wittenberg. However, 39 patients had to be excluded because of renal rest function with general low ultrafiltration rates (N = 37) or a lack of willingness (N = 2). Inclusion criteria of HD patients included age > 18 years and a history of haemodialysis treatment >12 weeks. Subjects with active malignancy, active infections and neurological disorders were excluded from the study. All patients gave their informed consent. Reasons for kidney failure included vascular nephropathy (23.8%), diabetic nephropathy (14.8%), glomerulonephritis (19.0%), interstitial nephritis (7.1%), autosomal-dominant kidney disease (14.8%) and others (21.4%). The fluid volume status was determined by bioelectrical impedance analysis (Nutriguard-MS, Data Input GmbH, Pöcking, Germany) after the last dialysis session of the week (Fridays or Saturdays), before the long interdialytic interval. Normohydration was defined as a range of impedance vectors falling within the reference 75% tolerance interval. Overhydration was defined by a Piccoli vector diagram > 75th percentile. Regarding the dialysis modalities, we found 11 patients dialysed with permanent dialysis catheters and 30 by arteriovenous fistula, while 1 patient had an arteriovenous graft. Forty-one patients were treated by high-flux dialysers, and one by a low-flux dialyser. All immunobiological samples from HD patients were taken after the long intradialytic interval and before start of the first dialysis session of the week. Blood samples were collected in the morning. The study was conducted according to the Declaration of Helsinki. Written informed consent was obtained from all study subjects, and the study protocol was approved by the local ethics committee.

### 4.2. Absolute Cell Count Determination

Trucount tubes (BD Biosciences, Heidelberg, Germany) were used to determine absolute cell count numbers in 50 µL of whole blood. Total leucocytes were identified by anti-CD45 (Miltenyi Biotec, Bergisch-Gladbach, Germany), granulocytes by anti-CD15 (Thermo Fisher Scientific, Darmstadt, Germany), monocytes by anti-CD14 (Milentyi Biotec) and lymphocytes by anti-CD3 (Miltenyi Biotec), -CD4 (Biolegend, Koblenz, Germany) and -CD8 staining (Miltenyi Biotec).

### 4.3. RNA Analysis Isolated from TEMPUS Tubes

For the detection of caspase-1 and cytokine transcripts, whole blood was drawn in TEMPUS™ Blood RNA Tubes (Life Technologies). Samples were stored at −80 °C as recommended, and analysis was performed after collection of all blood samples.

RNA was isolated using the TempusTM Spin RNA Isolation Reagent Kit (Life Technologies, Darmstadt, Germany). The RNA concentration and quality (260/280 ratio: 2.05 ± 0.005) were tested by the NanoDrop technique (PEQLAB Biotechnologie GmbH, Erlangen, Germany). Equal amounts of RNA (500 ng) were reverse transcribed using the FastGene Scriptase Basic cDNA Kit (Nippon Genetics Europe, Düren, Germany).

TNFAIP3 (A20, Hs01568117_m1), TNFAIP5 (Pentraxin3, Hs00173615_m1), TNFAIP8L2 (TIPE2, Hs01934946_s1), OTUD1 (Hs02596821_s), IL-10 (Hs00961622_m1), IL-22 (Hs01574154_m1), IL-26 (Hs00218189_m1) and ACTB (Hs01060665_m1) mRNA expression were analysed using TaqMan probes (Life Technologies) using qPCRBIO Probe Mix High-ROX (Nippon). The samples were processed in duplicate on a StepOnePlus Cycler (Life Technologies). Data were normalised by actin B and related to healthy control donor RNA. Thus, results are expressed as x-fold difference (RQ) compared to the healthy controls.

### 4.4. Sequence Analysis of Hyper- and Normovolemic Whole Blood Samples (mRNA-Seq)

Enough high-quality RNA for sequencing was available in 10 HD patients (7 normovolemic, 3 hypervolemic). Total RNA was isolated as described (Section 2.3.). The RNA integrity number (RNA) was determined using an Agilent 2100 Bioanalyzer (Thermo Fisher Scientific). The RIN value was >8 in all samples. QC/Library preparation, sequencing and analysis was performed by BMKGENE (Biomarker technologies (BMK) GmbH, Münster, Germany). As a reference database, “ftp://ftp.ensembl.org/pub/release-95/fasta/homo_sapiens/dna/” was used. Sequence-derived FKPM values (fragments per kilobase per million mapped fragments) were statistically analysed, finding their means ± SD, followed by Mann–Whitney analysis.

### 4.5. PBMC Isolation

PBMCs were isolated by Ficoll density centrifugation (GE Healthcare, Solingen, Germany) from EDTA blood samples drawn from the dialysis access before the dialysis session. The quality of isolated cells was tested by 7-AAD staining. The vitality of the PBMCs was 99.3% ± 0.7 for CO, 99.7 ± 0.3 for LD and 99.5 ± 0.5.

### 4.6. Treatment of PBMCs

A total of 0.8–1.0 × 10^7^ cells per patient were incubated in 25 cm^3^ tissue culture flasks (TPP, Trasadingen, Switzerland). After 16 h under 5% CO_2_ at 37 °C, the flasks were treated with cycloheximide (CHX; 100 µg/mL, Sigma-Aldrich, Steinheim, Germany) for 15 min.

### 4.7. Preparation of Cytosolic Lysates

After scraping out the tissue flasks, the cells were washed using ice-cold PBS. Afterwards cells were lysed on ice for 30 min using a cycloheximide-containing lysis buffer (5 mM Tris-Base, 1.25 mM MgCl_2_, 1.5 mM KCl, 0.5% Sodium-deoxycholate (Sigma-Aldrich), 2 mM DTT (Roth, Karlsruhe, Germany), 0.5% Triton-X-100 (Roth), RNase-out (Thermo-Fisher Scientific, Darmstadt, Germany) and 100 µg/mL CHX). Prior to use, the lysis buffer was supplemented with 6 µL/mL of RNaseOut (ThermoFisherScientific). An aliquot of the lysed suspension was taken for RNA extraction (mRNA sample before gradient preparation).

### 4.8. Preparation of Linear Sucrose Gradients

We prepared 5 and 45% sucrose solutions in polysome buffer (5 mM Tris-Base, 1.25 mM MgCl_2_, 1.5 mM KCl, RNase-out and 100 µg/mL CHX); a layering device (BioComp, ScienceServices GmbH, München, Germany) was used to fill the open-top polyclear ultracentrifuge tubes (13 × 51 mm, Seton Scientific, München, Germany) with the sucrose solutions, followed by linear gradient formation using the gradient maker (BioComp). Afterwards the samples were applied to the corresponding tubes. The tubes were sealed with rate zonal caps (BioComp), placed in a precooled MLS 50 rotor (Beckman Coulter, Krefeld, Germany) and centrifuged at 130,000× *g* for 1.5 h. Afterwards, 430 µL fractions were collected with the Piston Gradient Fractionator (BioComp). The process was UV-monitored (integrated UA6 UV monitor) using BioComp software version 2.10.00. Fraction 5 was identified as “the monosomal fraction”, followed by fraction 6 which was determined as the “early polysomal fraction”; fraction 9 was defined as the “late polysomal fraction.”

### 4.9. RNA/cDNA/qPCR from Profiling Experiments

RNA was isolated from PBMC lysates using the Direct-zol MiniPrep Plus Isolation Kit (ZymoResearch, Freiburg, Germany). The RNA concentration and quality of the different fractions were tested by Z-Tecan Infine^®^200 Pro technique (Tecan Group, Männedorf, Germany).

Equal amounts of RNA (45 ng) were reverse transcribed using the High-Capacity cDNA Reverse Transcription Kit (Thermo Fisher Scientific, Darmstadt, Germany).

IL-10 (Hs00961622_m1), TNFAIP8L2 (TIPE2, Hs01934946_s1), OTUD1 (Hs02596821_s) and RPL37A (Hs01102345_m1) mRNA expression were analysed using TaqMan probes (Thermo Fisher Scientific) and the qPCRBIO Probe Mix High-ROX (Nippon Genetics, Düren, Germany). The samples were processed in duplicate on a StepOnePlus Cycler (Thermo Fisher Scientific). Data were normalised by RPL37A (dCt method).

### 4.10. Cytokine Analysis

CRP was analysed in the sera using ELISA techniques (CRP, Biomol, Hamburg, Germany). Data were analysed on an ELX808 microplate reader (Bio-Tek Inc., Berlin, Germany).

### 4.11. Statistics

Results are expressed as means ± SD. All continuous variables were controlled for normal distribution using the D’Agostino–Pearson omnibus test. Continuous data were compared by the T-test, Mann–Whitney test or by one-way ANOVA followed by Friedman’s or Tukey’s post-test as appropriate. Categorical variables were analysed by the chi-square test. The relation of monosomal OTUD1 mRNA expression to putative confounders was analysed using a multivariate linear regression model. The model contained OTUD1 mRNA expression data (monosomal fraction 5) and phase angle as fixed factors, and age, Kt/V and CRP values as covariates. All calculations were carried out using SPSS 21.0 (SPSS Inc., Chicago, IL, USA) or GraphPad Prism 9.2.0 statistics software (GraphPad Software Inc., La Jolla, CA, USA). The level of significance was set at *p* < 0.05.

## 5. Conclusions

This study confirms that the anti-inflammatory repertoire of hypervolemic HD patients is insufficiently developed to achieve cellular homeostasis. IL-10 and TIPE2 cannot balance uremic-toxin- and hypervolemia-induced inflammation. The action of OTUD1 is multifactorial, and this study could not determine which of the different molecules, including transcription factors, are targets of OTUD1. TIPE2, a putative target of OTUD1, is not differently expressed between N and H patients. Therefore, we suggest that OTUD1 does not differently regulate TIPE2 among normo- and hypervolemic patients.

## Figures and Tables

**Figure 1 ijms-26-08922-f001:**
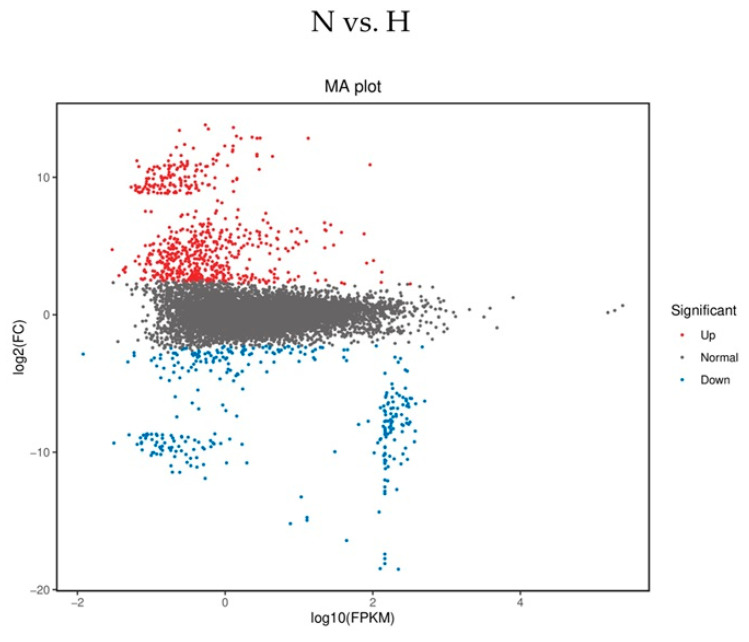
Exemplary dot plot of differential gene expression in a normovolemic (N) versus a hypervolemic HD patient (H). Depicted is an MA plot with log2 fold changes (M) versus the average expression signal (A) of a normovolemic (N) versus a hypervolemic HD patient (H), who differ in their phase angle. The red dots represent upregulated and blue dots downregulated genes. Similar expression levels are represented by black dots.

**Figure 2 ijms-26-08922-f002:**
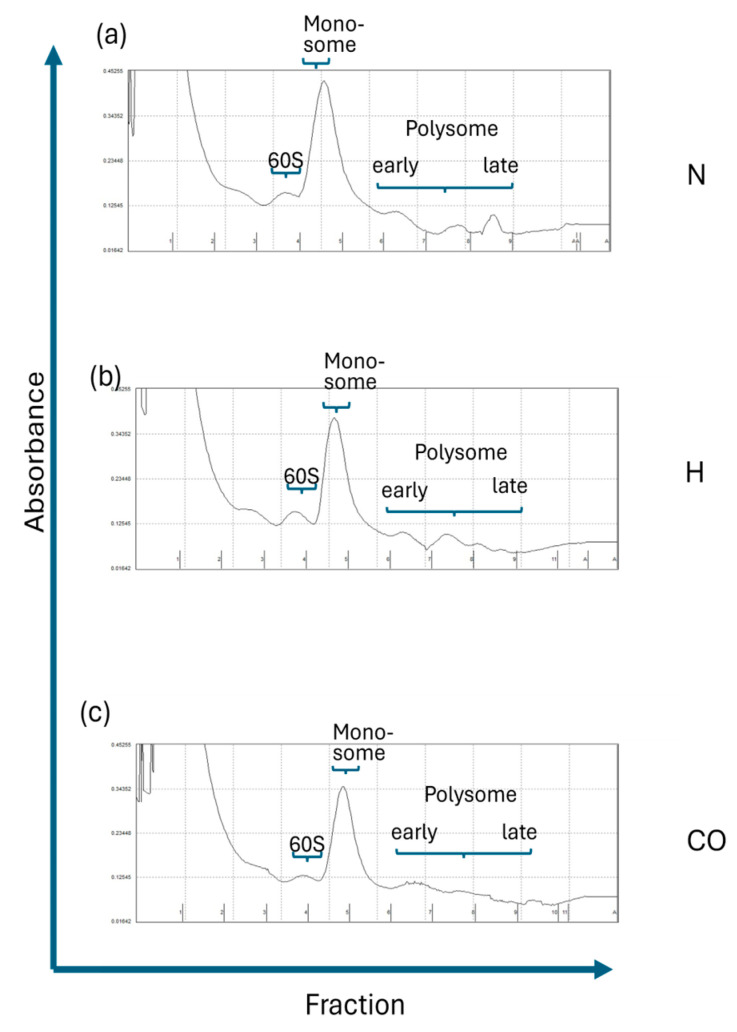
Representative presentation of fractionated samples of isolated mono- and polysomes of a normovolemic, a hypervolemic and a healthy control subject. The positions of the 60S, monosomal (fraction 5) and polysomal peaks (fractions 6–9) are given.

**Figure 3 ijms-26-08922-f003:**
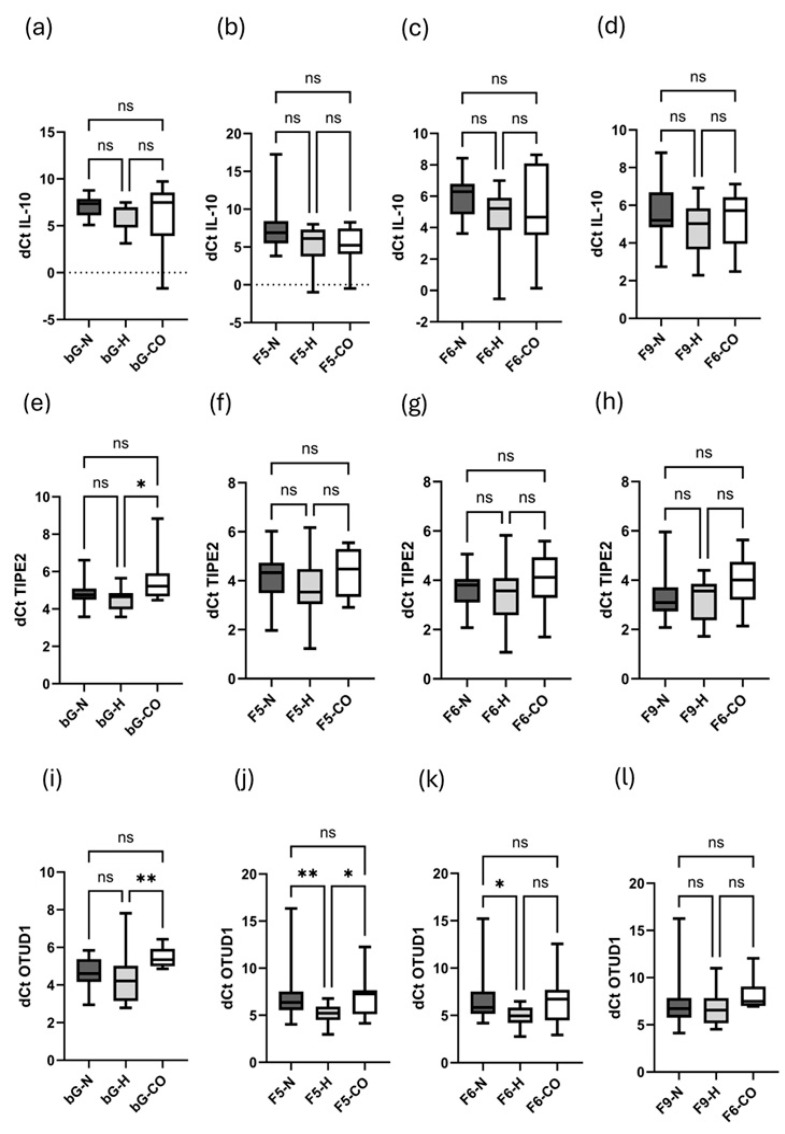
Transcriptional (bG, before gradient) and translational activities. The monosomal [5] and early [6] and late polysomal [9] activities of IL-10, TIPE2 and OTUD1 in normovolemic (N), hypervolemic (H) and healthy (CO) subjects are depicted. (**a**–**d**) represent translational activities of IL-10 [(**a**) data under basal conditions, before gradient; (**b**) monosomal translational activities (F5); (**c**) early translational activities (F6); (**d**) late translational activities (F9)]. (**e**–**h**) represent translational activities of TIPE2 [(**e**) data under basal conditions, before gradient; (**f**) monosomal translational activities (F5); (**g**) early translational activities (F6); (**h**) late translational activities (F9)]. (**i**–**l**) represent translational activities of OTUD1 [(**i**) data under basal conditions, before gradient; (**j**) monosomal translational activities (F5); (**k**) early translational activities (F6); (**l**) late translational activities (F9)]. Data are presented as box plots with 25th and 75th percentiles. * *p* < 0.05, ** *p* < 0.01, ns: not significant.

**Table 1 ijms-26-08922-t001:** Basic characteristics of the study groups.

	N (n = 25)	H (n = 17)	*p*-Value N vs. H	CO (n = 9)	*p*-Value CO vs. N vs. H
Age (years)	55.6 ± 13.9	65.6 ± 14.9	0.030	39.7 ± 18.4	0.005 H vs. CO
Gender (f, %)	52.0	29.4	0.208	33.0	0.306
BMI (kg/m^2^)	26.0 ± 5.3	29.3 ± 9.0	0.143	-	
Diabetes (%)	20.0	41.2	0.174	-	
Epo (I.U./week)	8850 ± 4904	12,157 ± 7753	0.097	-	
Phase angle (°)	5.4 ± 1.0	4.1 ± 0.9	0.001	-	
Resistance	629.1 ± 111.2	449.7 ± 62.5	0.001	-	
Reactance	59.0 ± 13.6	32.4 ± 8.1	0.001	-	
Total body water (L)	37.4 ± 9.3	47.6 ± 8.7	0.001	-	
BCM (Kg/m^2^)	25.0 ± 7.8	26.5 ± 7.1	0.523	-	
ICW (L)	22.9 ± 4.4	26.1 ± 3.9	0.019	-	
ECW (L)	14.5 ± 5.0	21.4 ± 5.6	0.001	-	
Sys. RR after HD (mmHg)	131.9 ± 19.5	148.9 ± 25.9	0.020	-	
Dia. RR after HD (mmHg)	75.4 ± 15.5	70.8 ± 13.2	0.320	-	
PP after HD (bpm)	71.8 ± 10.9	66.5 ± 11.5	0.082	-	
Kt/V	1.5 ± 0.4	1.3 ± 0.2	0.026	-	
Creatinine (µmol/L)	921.9 ± 218.5	667.8 ± 206.8	0.001	-	
Hb (mmol/L)	6.9 ± 0.4	6.7 ± 0.8	0.288	-	
CRP (mg/dL)	18.3 ± 11.8	29.9 ± 17.4	0.024	0.9 ± 0.6	0.001
Albumin (g/L)	40.6 ± 3.3	38.9 ± 3.5	0.108	-	
Bicarbonate (mmol/L)	22.6 ± 1.5	24.2 ± 2.0	0.004	-	
Sodium (mmol/L)	138.1 ± 3.3	138.8 ± 3.5	0.527	-	
Potassium (mmol/L)	5.7 ± 0.7	5.5 ± 0.8	0.377	-	
Calcium (mmol/L)	2.2 ± 0.2	2.3 ± 0.2	0.615	-	
Phosphate (mmol/L)	2.0 ± 0.4	1.7 ± 0.4	0.019	-	

Abbreviations: BMI: body mass index; Epo: erythropoietin; BCM: body cell mass; ICW: intracellular water; ECW: extracellular water; sys. RR: systolic blood pressure; dia. RR: diastolic blood pressure; PP: pulse pressure; Kt/V: dialyser clearance of urea per dialysis time; CRP: C-reactive protein. Data is presented as means ± SD. Statistical analysis was performed by the Mann–Whitney test or by one-way ANOVA.

**Table 2 ijms-26-08922-t002:** Leucocyte cell numbers in normovolemic (N) and hypervolemic (H) haemodialysis patients and healthy controls.

	N (n = 25)	H (n = 17)	CO (n = 9)	*p*-Value N vs. H vs. CO
CD45+ Leucocytes (10^6^/mL)	9.7 ± 2.8	9.5 ± 3.2	9.3 ± 2.4	0.089
CD15+ Granulocytes (10^6^/mL)	5.5 ± 2.2	5.8 ± 2.2	4.2 ± 1.1	0.191
CD3+ Lymphocytes (10^6^/mL)	1.3 ± 0.36	1.2 ± 0.5	1.6 ± 0.4	0.086
CD14+ Monocytes (10^6^/mL)	0.7 ± 0.3	0.7 ± 0.2	0.6 ± 0.1	0.396
CD4+ Lymphocytes (10^6^/mL)	0.8 ± 0.4	0.6 ± 0.2	0.9 ± 0.4	0.047
CD8+ Lymphocytes (10^6^/mL)	0.4 ± 0.3	0.3 ± 0.2	0.5 ± 0.3	0.079
CD19+ B-cells (10^6^/mL)	0.3 ± 0.2	0.2 ± 0.1	0.3 ± 0.3	0.553

Data is presented as means ± SD. Statistical analysis was performed by one-way ANOVA.

**Table 3 ijms-26-08922-t003:** Differentially expressed genes (DEGs) detected by mRNA-seq analysis.

Downregulation in N			
Symbol	FKPM (N)	FKPM (H)	*p*-Value
ASPRV1	2.94 × 10^6^	1.01 × 10^7^	1.00 × 10^−2^
CTSE	4.51 × 10^0^	1.86 × 10^6^	4.02 × 10^−2^
EXOC4	4.78 × 10^7^	1.36 × 10^7^	1.37 × 10^−2^
GPLD1	2.55 × 10^0^	1.20 × 10^6^	1.19 × 10^−2^
HBG2	6.90 × 10^8^	2.01 × 10^9^	3.31 × 10^−2^
HLA-DRB5	4.35 × 10^7^	1.01 × 10^8^	1.96 × 10^−3^
IGKV2D-28	5.48 × 10^6^	1.08 × 10^7^	3.77 × 10^−2^
MYBL2	4.64 × 10^0^	1.54 × 10^6^	1.62 × 10^−2^
MYL6B	1.32 × 10^5^	2.43 × 10^6^	5.59 × 10^−3^
OTUD1	2.22 × 10^0^	2.02 × 10^6^	3.98 × 10^−3^
Upregulation in N			
ABCA3	4.15 × 10^8^	1.02 × 10^0^	1.70 × 10^−4^
ANO5	2.03 × 10^0^	1.27 × 10^−1^	1.25 × 10^−4^
AXIN1	2.92 × 10^9^	1.14 × 10^7^	1.04 × 10^−4^
CAPN15	2.00 × 10^9^	8.41 × 10^6^	1.15 × 10^−4^
CCDC154	3.68 × 10^8^	6.77 × 10^−1^	1.38 × 10^−4^
CCNF	9.31 × 10^7^	1.00 × 10^0^	2.05 × 10^−4^
CIAO3	2.44 × 10^9^	2.07 × 10^6^	1.02 × 10^−4^
FAM173A	1.67 × 10^9^	8.09 × 10^6^	2.06 × 10^−4^
MCRIP2	1.54 × 10^9^	4.36 × 10^6^	1.14 × 10^−4^
SOX8	1.10 × 10^9^	3.02 × 10^−1^	1.01 × 10^−4^
TNFAIP8L2	1.05 × 10^8^	3.07 × 10^7^	4.92 × 10^−2^

Abbreviation: FKPM (fragments per kilobase). Data is presented as means ± SD. Statistical analysis was performed by the Mann–Whitney test.

**Table 4 ijms-26-08922-t004:** Gene expression of some TNFAIP and IL-10 family members among normo- (N) versus hypervolemic (H) patients, as analysed by mRNA-seq analysis.

Symbol	FKPM (N)	FKPM (H)	*p*-Value
TNFAIP3 (A20)	4.538 × 10^7^	9.224 × 10^6^	0.278
TNFAIP5 (Pentraxin3)	3.871 × 10^0^	1.410 × 10^0^	0.443
TNFAIP8L2 (TIPE2)	1.05 × 10^8^	3.07 × 10^7^	0.049
IL-10	n.d.	n.d.	
IL-22	n.d.	n.d.	
IL-26	n.d.	n.d.	

Abbreviation: FKPM (fragments per kilobase million); n.d.: not detected. Data is presented as means ± SD. Statistical analysis was performed by the Mann–Whitney test.

**Table 5 ijms-26-08922-t005:** mRNA expression of some TNFAIP and IL-10 family members and OTUD1.

Symbol	(N)	(H)	*p*-Value
TNFAIP3 (A20)	1.7 ± 0.9	1.4 ± 0.6	0.397
TNFAIP5 (Pentraxin3)	1.5 ± 1.0	1.9 ± 1.2	0.451
TNFAIP8L2	1.5 ± 0.7	1.0 ± 0.2	0.006
IL-10	1.5 ± 0.9	2.0 ± 1.2	0.198
IL-22	2.4 ± 1.4	2.9 ± 1.1	0.194
IL-26	0.8 ± 0.6	1.3 ± 1.2	0.350
OTUD1	1.5 ± 0.6	1.3 ± 0.2	0.295

Data is presented as means ± SD. Statistical analysis was performed by the Mann–Whitney test or by one-way ANOVA.

**Table 6 ijms-26-08922-t006:** Multivariate linear regression model.

Dependent Variable	Parameter	Regression Coefficient B	Standard Error	T	Significance	95% Confidence Interval		Partial Eta-Quadrat
						Lower Limit	Upper Limit	
OTUD1 mRNA (F5—monosomal)	Constant	7.367	2.469	2.983	0.005	2.337	12.397	0.218
	Kt/V	0.850	1.297	0.655	0.517	−1.791	3.491	0.013
	CRP	−0.081	0.030	−2.657	0.012	−0.142	−0.019	0.181
	Age	−0.004	0.030	−0.145	0.886	−0.066	0.057	0.001
Phase angle (°)	Constant	6.451	0.891	7.245	0.000	4.638	8.265	0.621
	Kt/V	0.509	0.468	1.089	0.284	−0.443	1.462	0.036
	CRP	−0.023	0.011	−2.127	0.041	−0.046	−0.001	0.124
	Age	−0.029	0.011	−2.630	0.013	−0.051	−0.006	0.178

Using this model, we see that monosomal OTUD1 mRNA expression is only dependent on the inflammatory marker CRP and independent of dialysis quality (Kt/V) and age. In contrast, hydration status (phase angle) is dependent on age and CRP.

## Data Availability

Data is contained within this article.

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
