# Peer review of "Different Translational Activities of Inflammatory Regulators Associated with Hypervolemia in Haemodialysis Patients"

_ijms, 2025, doi:10.3390/ijms26188922_

Round 1

Reviewer 1 Report

Comments and Suggestions for Authors

In the present article, titled "Different Translational Activities of Inflammatory Regulators Associated with Hypervolemia in Hemodialysis Patients" by Ulrich et al., the authors aim to study the translational profile of IL-10, TIPE2, and OTUD1, possibly triggered by fluid overload in HD patients. I believe the topic is very interesting; however, I have some comments that warrant consideration and could help improve the manuscript before it is accepted.

Specific Comments:

  • The criteria used to define hypervolemia and normovolemia should be explicitly stated. The Nutriguard-MS was used for BIA. What were the cutoff values for overhydration?
  • Were patients considered to be at clinical dry weight based on standard clinical criteria?
  • It is unclear when fluid status was assessed: was it before or after dialysis, and during the short or long interdialytic interval?
  • Additional objective parameters, such as lung ultrasound (B-lines) or inferior vena cava diameter, could strengthen the assessment of volume status. Were any of these used or considered?
  • The study population appears to have markedly low haemoglobin levels across the board. Was there a specific reason for this (e.g., limited use of erythropoiesis-stimulating agents, iron deficiency)? Given that target haemoglobin levels in HD patients are typically 100–110 g/L, this warrants further explanation.
  • RNA sequencing was conducted in only 10 patients. What was the rationale for this small subgroup? This limited sample size could introduce selection bias and limit the generalizability of the molecular findings. How were these 10 patients selected?
  • The manuscript would benefit from a clearer explanation of how the 42 HD patients were selected. Were these consecutive patients, or a specific subgroup within the dialysis centre?
  • Were the patients dialysed over an AVF or a dialysis catheter?
  • The observed differences in inflammatory profiles may be influenced by various confounding factors such as age, nutritional status (e.g., phase angle), dialysis adequacy (Kt/V), and others.
  • Was any multivariate or regression analysis performed to account for these potential confounders? If not, this should be considered, or at least acknowledged as a limitation.

Author Response

We would like to thank the reviewers for their helpful comments!

Response to Reviewer 1 Comments

Point 1:  The Reviewer wants to see the definition of normo- and hypervolemia being explicitly stated. Further on he asks for the cutoff values used for definition of overhydration.

Response 1:

We thank the Reviewer for these useful suggestions. We added the following sentences to the Material and Method section: “Normohydration was defined as a range of impedance vectors falling within the reference 75% tolerance interval. Overhydration was defined by the Piccoli vector diagram >75th percentile”. Material and Method section, lines 330-332.

Point 2: The Reviewer asks if dry weight of patients was part of the clinical examination

Response 2:  

The dry weight determination is of course part of the regular patient visits. Dry weight gives an idea how the putative ultrafiltration rate should be fixed. However, bioimpedance analysis in the multi-frequency mode is also recognized as a valuable tool for the evaluation of dry weight which additionally gives information about the nutrition and hydration state of HD patients.

Point 3: The Reviewer points out that it is unclear when fluid status was assessed: before or after dialysis and during the short and the long interdialytic interval.

Response 3:  

The Reviewer is right, this information is missing: We added the following sentence to the Material and Method section: “The fluid volume status was determined by bioelectrical impedance analysis (Nutriguard-MS, Data Input GmbH, Pöcking, Germany) after the last dialysis session of the week (Fridays or Saturdays), before the long interdialytic interval.”  Material and Method section, lines 327-331.

Point 4: The Reviewer is of the opinion that additional objective parameters such as lung ultrasound (B-lines) or inferior cava diameter could strengthen the assessment of volume status.

Response 4:

The Reviewer is right. Validation of results using different analytical methods is highly recommended. But alternative methods like lung ultrasound or inferior cava diameter assessments were not available for this study.

The following sentences were added to the section “Study limitations”: “We used bioimpedance measurement to determine the hydration status of our HD patients. Validation of these data by other methods such as lung ultrasound or inferior cava diameter assessments were unfortunately not available for this study”. Study limitations, lines 302-304.

Point 5: The Reviewer remarks that HD patients have markedly low haemoglobin levels (normal range 10-11 g/dL).

Response 5:

I think this is a misunderstanding. The Hb levels in our study are given in mmol/L. That means the average Hb value of 6.9 mmol/L in the N group corresponds to 11.12 g/dl and 6.7 mmol in the H group corresponds to 10.8 g/dl. These values are indeed characteristic for HD patients.

Point 6: The Reviewer remarks that RNA sequencing was conducted in only 10 patients. What was the rationale for this small subgroup? Further on, the limited sample size could introduce selection bias and limit the generalizability of molecular findings.

Response 6:

We confess that RNA-seq data of 10 patients are quite a low number to deduce any generalizability. Originally 20 samples comprising three ml of whole blood (10 N and 10 H patients) had been chosen for RNA-isolation to get the required RNA amounts for mRNA sequencing. Unfortunately, there was a technical problem during RNA isolation destroying many RNAs. So only 10 samples were left for sequencing.

We recognize this as a limitation and added the following information to the section “Study limitations”: “Furter on, for the RNA-sequence analysis only samples of 10 patients were available. This indeed could bias the results and may limit the generalizability of the molecular findings”. Study limitations, lines 304-307.

Point 7: The Reviewer suggests giving a clearer explanation of how the 42 HD patients have been selected

Response 7:

We thank the Reviewer for this suggestion and added the following sentences to the Material and Method section: At first all 81 HD patients have been recruited from the KFH dialysis centre which is associated with the Department of Internal Medicine II of the University Halle-Wittenberg. However, 39 patients had to be excluded because of a satisfied renal rest function with general low ultrafiltration rates (N=37) or lack of willingness (N=2). Material and Method section (5.1 Study population). Material and Method section, lines 318-321.

Point 8: The Reviewer wants to know whether dialysis patients were dialysed by arteriovenous fistula or dialysis catheter?

Response 8:

The Reviewer is right. Some basic information about dialysis modalities were missing. We added the following sentences to the Material and Method section: “Regarding the dialysis modalities we found 11 patients dialysed with permanent dialysis catheters, 30 by arteriovenous fistulas, 1 patient had an arteriovenous graft. Forty-one patients were treated by high-flux, 1 by low-flux dialyzers. Material and Method section, Lines 332-334.

Point 9: The Reviewer remarks that the observed differences in inflammatory profiles may be influenced by various confounding factors such as age, phase angle, Kt/V and CRP. If the potential confounder is not analysed by multivariate analysis the influence on study results should be acknowledged as limitation.

Response 9:

We thank the Reviewer for his suggestion. It is very likely that the fractional OTUD1 expression is influenced by confounders. Approaching this question, we performed a multivariate linear regression analysis using OTUD1 mRNA expression data (monosomal (fraction 5) which is significantly different between N and H patients) and phase angle (determinant of cell integrity and the nutritional status) as fixed factors and age, Kt/V and CRP-values as covariates.

The following sentences were added to the Statistic section: “The relation of monosomal OTUD1 mRNA expression to putative confounders was analysed using a multivariate linear regression model. The model contained OTUD1 mRNA expression data (monosomal fraction 5) and phase angle as fixed factors and age, Kt/V and CRP-values as covariates”. Statistics, lines 436-439

Multivariate linear regression model

Dependent Variable

Parameter

Regression-

Coefficient B

Standard

error

T

Significance

95%-Confidence interval

Partial

 Eta-Quadrat

Lower

limit

Upper

limit

OTUD1 mRNA

 (F5-monosomal)

Constant

7,367

2,469

2,983

,005

2,337

12,397

,218

Kt/V

,850

1,297

,655

,517

-1,791

3,491

,013

CRP

-,081

,030

-2,657

,012

-,142

-,019

,181

Age

-,004

,030

-,145

,886

-,066

,057

,001

Phase angle (°)

Constant

6,451

,891

7,245

,000

4,638

8,265

,621

Kt/V

,509

,468

1,089

,284

-,443

1,462

,036

CRP

-,023

,011

-2,127

,041

-,046

-,001

,124

Age

-,029

,011

-2,630

,013

-,051

-,006

,178

The following sentences were added to the Result section:

2.6 Multivariate linear regression model

“There remains the question whether the monosomal difference in OTUD1 mRNA expression between N and H patients is dependent from other relevant factors. Approaching this question, we performed a multivariate linear regression analysis using OTUD1 mRNA expression data (monosomal (fraction 5) which is significantly different between N and H patients) and phase angle (determinant of cell integrity and nutritional status) as fixed factors and age, Kt/V and CRP-values as covariates”. Result section, lines 194-203.

Using this model, we see that monosomal OTUD1 mRNA expression is only dependent on the inflammatory marker CRP and independent of dialysis quality (Kt/V) and age. In contrast, the nutritional status (phase angle) is dependent on age and CRP.

This information was added to the result section. Result section, lines 204-206.

The following sentences were added to the Discussion section:

“Most interestingly N and H patients significantly differed about age. As age is a powerful predictor of most diseases this fact may impact our study setup, i.e. phase angle, IL-10 and OTUD1 expression. However as analysed by multivariate expression age rather affects the phase angle than OTUD1 mRNA expression. The phase angle reflects cellular health and the nutritional status. Thus, low phase angle values coincide with low cellular integrity and a worse nutritional status. This appears to make sense for “the older” H patients. In contrast OTUD1 mRNA expression is dependent on the inflammatory status but not of age or Kt/V”. Discussion section, lines 215-222.

Reviewer 2 Report

Comments and Suggestions for Authors

In general, I consider this manuscript is very clear and interesting, but I have some recommendations:

  1. It is well known that IL10 is an antiinflammatory gene related with the increase of expression and activity of antioxidant enzymes (SOD, CAT, GPx, GR). Considering that inflammation is not conclusive, Have you considered including in the discussion or performing measurements related to oxidative stress? It will likely reinforce the theory or open a new perspective.
  2. Why do you consider patients in HD? In general inflammatory profile is not the best

Author Response

Response to Reviewer 2 Comments

Point 1: The Reviewer states that IL-10 is related to the increase of expression and activity of antioxidant enzymes (SOD, CAT, GPx, GR). This association should be considered and at least being discussed.

Response 1:

The Reviewer is right. Oxidative stress may have a high impact on the inflammatory regulators studied in our HD patients. The following sentences were added to the Discussion section:

“Regarding other mechanisms which may influence inflammatory regulators is oxidative stress.  It is well known that hemodialysis is associated with increased oxidative stress which cannot adequately be compensated for antioxidative capacity (superoxide dismutase) of dialysis patients [36, 37]. Further on, there is evidence that, for example serum protein carbonyl, also a marker of oxidative stress, too, is elevated in overhydrated HD patients [38]. Unfortunately, we did not measure oxidative stress parameters to see if there are any relations IL-10, OTUD1 or TIPE2. So, it is quite possible that oxidative stress affects the anti-inflammatory machinery most probably IL-10 and OTUD1. This, however, is a future topic to be studied”. Discussion section, lines 289-298.

These References were added to the Reference list:

  1. Rysz, J.; Franczyk, B.; Ławiński, J.; Gluba-Brzózka, A. Oxidative Stress in ESRD Patients on Dialysis and the Risk of Cardiovascular Diseases. Antioxidants (Basel) 2020, 9, doi:10.1053/j.ajkd.2005.04.031.

  1. Karamouzis, I.; Sarafidis, P.A.; Karamouzis, M.; Iliadis, S.; Haidich, A.-B.; Sioulis, A.; Triantos, A.; Vavatsi-Christaki, N.; Grekas, D.M. Increase in oxidative stress but not in antioxidant capacity with advancing stages of chronic kidney disease. Am. J. Nephrol. 2008, 28, 397–404, doi:10.1159/000112413.

  1. Song, Y.R.; Kim, J.-K.; Lee, H.-S.; Kim, S.G.; Choi, E.-K. Serum levels of protein carbonyl, a marker of oxidative stress, are associated with overhydration, sarcopenia and mortality in hemodialysis patients. BMC Nephrol. 2020, 21, 281, doi:10.1186/s12882-020-01937-z.

Point 2: The Reviewer wants to know whether dialysis patients – although not having a high inflammatory profile – were chosen as study subjects.

Response 2:

 Inflammation is a characteristic feature of hemodialysis patients and although this kind of inflammation appears to be low (termed microinflammation with CRP values <50 mg/dl) it is detrimental for the patients for microinflammation is permanently present.  We are of the opinion that the patho-mechanism behind the low-level inflammatory HD scenery is a very important issue to study.

Round 2

Reviewer 1 Report

Comments and Suggestions for Authors

The authors have sufficiently responded to all reviewer comments.